# Does the Potocki–Lupski Syndrome Convey the Autism Spectrum Disorder Phenotype? Case Report and Scoping Review

**DOI:** 10.3390/jpm13030439

**Published:** 2023-02-28

**Authors:** Oksana I. Talantseva, Galina V. Portnova, Raisa S. Romanova, Daria A. Martynova, Olga V. Sysoeva, Elena L. Grigorenko

**Affiliations:** 1Center for Cognitive Sciences, Sirius University of Science and Technology, 354340 Sirius, Russia; 2Institute of Higher Nervous Activity and Neurophysiology, Russian Academy of Sciences, 117485 Moscow, Russia; 3Department of Psychology, University of Houston, Houston, TX 77204, USA; 4Department of Molecular and Human Genetics, Baylor College of Medicine, Houston, TX 77030, USA; 5Child Study Center, Yale University, New Haven, CT 06519, USA

**Keywords:** Potocki–Lupski syndrome, 17p11.2, PTLS, autism, ASD, EEG, language, speech

## Abstract

Potocki–Lupski Syndrome (PTLS) is a rare condition associated with a duplication of 17p11.2 that may underlie a wide range of congenital abnormalities and heterogeneous behavioral phenotypes. Along with developmental delay and intellectual disability, autism-specific traits are often reported to be the most common among patients with PTLS. To contribute to the discussion of the role of autism spectrum disorder (ASD) in the PTLS phenotype, we present a case of a female adolescent with a de novo dup(17) (p11.2p11.2) without ASD features, focusing on in-depth clinical, behavioral, and electrophysiological (EEG) evaluations. Among EEG features, we found the atypical peak–slow wave patterns and a unique saw-like sharp wave of 13 Hz that was not previously described in any other patient. The power spectral density of the resting state EEG was typical in our patient with only the values of non-linear EEG dynamics: Hjorth complexity and fractal dimension were drastically attenuated compared with the patient’s neurotypical peers. Here we also summarize results from previously published reports of PTLS that point to the approximately 21% occurrence of ASD in PTLS that might be biased, taking into account methodological limitations. More consistent among PTLS patients were intellectual disability and speech and language disorders.

## 1. Introduction

The Potocki–Lupski syndrome (PTLS; OMIM: 610883) is a relatively newly found genetic disorder resulting from interstitial duplication of chromosome 17 band p11.2, usually with a length of 3.7 Mb with a predicted incidence of approximately 1 in 25,000 live births [1]. PTLS may underlie a wide range of congenital abnormalities, including mild dysmorphic features (such as pronounced nose, triangular or square face, down-slanting palpebral fissures, and micrognathia), poor feeding, failure-to-thrive in infancy, obstructive and central sleep apnea, history of seizures, microcephaly, ophthalmic, orthopedic, cardiovascular, oropharyngeal, and renal anomalies [2]. Similarly, the range of behavior disturbances and neurodevelopmental disorders is also heterogeneous. The most common symptoms include a history of developmental delay, borderline to severe intellectual disability (ID), speech and language disorders, lack of executive functions and aggressivity, anxiety, withdrawal, and features of attention-deficit/hyperactivity disorder (ADHD), and autism spectrum disorder (ASD) [2,3,4]. Regarding ASD in the first relatively large-scale studies, searching PTLS phenotype, it was proposed that, while ASD is a non-absolute yet common feature of PTLS, 17p11.2 could be strongly considered as a new region implicated in the genetics of ASD [1]. However, in the following research, the role of autistic features in the phenotype of PTLS has been questioned [5]. Thus, according to the most recent review, the prevalence of ASD among patients with PTLS comprised a more modest proportion of the comorbidity equal to 37.9% [2] compared with the results of the study by Treadwell-Deering et al. [1], where the prevalence of ASD was about 80% (on the sample of 15 patients with PTLS). To contribute to the discussion of the role of ASD in the PTLS phenotype, here we report a case of a 13-year-old Russian female child with confirmed de novo duplication 17p11.2 [6], focusing on in-depth clinical, behavioral, and electrophysiological assessments, and summarize related characteristics from existing literature updating the previous review [2].

## 2. Materials and Methods

### 2.1. Clinical and Behavioral Assessment

To conduct a comprehensive evaluation of the patient in the domains of speech, language, intelligence, and adaptive functioning, we used a clinical interview and the following battery of standardized tools: Russian adaptation [7,8] of the Preschool Language Scales, Fifth Edition (PLS-5) [9]; the Assessment of the Development of Russian (ORRIA) [10]; the Universal Nonverbal Intelligence Test, Second Edition (UNIT-2) [11]; and the Russian adaptation [12] of the Vineland Adaptive Behavior Scales, Second Edition (Vineland-II) [13]. Given that the Russian version of the PLS-5 (RPLS-5) and ORRIA are not normed yet and that both instruments are not appropriate for the patient’s biological age, we provided only descriptive results for these tools. To evaluate key manifestations of ASD, two methods, whose combination is reported to be the closest to the gold standard of ASD diagnostics [14], were employed: the Autism Diagnostic Interview–Revised (ADI-R) [15] and the Autism Diagnostic Observation Schedule (ADOS-2) [16]. For both methods, the Russian-adapted versions were used for the evaluation [17,18], and both were administered by a trained clinical psychologist (OT). The final diagnostic decision was based on the discussion by the research team according to DSM-5 TR criteria [19].

### 2.2. Electroencephalographic Assessment

To investigate alterations in the patient’s electroencephalogram (EEG), we used EEG data from 37 healthy controls from 12 to 15 years from another research project that were recorded according to the same protocol.

#### 2.2.1. EEG Recording

EEG data were recorded using a 28-channel NeuroTravel (Firenze, Italy) system with connected earlobe electrodes used as a reference and the grounding electrode placed centrally. Electrodes were arranged according to the international 10–10 system. EEG registration was conducted in awake patients with open eyes during the daytime and lasted for 1756 s. The signal was sampled at 500 Hz and filtered with an online bandpass filter of 0.016–70 Hz and with a notch filter at 50 Hz. The electrode impedances were below 10 kΩ.

#### 2.2.2. EEG Analysis

We analyzed EEG fragment 1756 s of eyes open condition for the patient and 1550–2000 s of eyes open condition for each child of the control group (1803 ± 79 s). Independent component analysis (ICA) was used when needed to subtract the most evident artifacts [20]. The three separate neurologists (including GP) with expert certification identified and interpreted the EEG data, reaching common decisions.

The process of EEG was videotaped to check typical clinical events or seizures. 

The following phenomena were analyzed:
The presence and the coverage of diffuse rhythmic activity or the generalized background slowing.Epileptiform EEG abnormalities:
Sporadic wave discharges, spikes, and multi-spikes are classified as a benign focal epileptiform discharge of childhood without clinical correlates.Episodic peak–wave or slow spike–wave complexes, which were not accompanied by clinical events and did not show repetitive structure, generalization, or secondary generalization, which correlated (or not) with clinical events. The topography of this activity was also taken into account.Typical or atypical epileptiform discharges manifesting with secondary generalized spike–slow wave discharges or spike–wave discharges that correlated (or not) with clinical events.

To compare the parameters of the EEG between the patient and healthy controls, we used the 1, 2, 3, and 4 sigmas measured using data from nine electrodes (F3, Fz, F4, C3, Cz, C4, P3, Pz, P4) averaged for each subject.

The power spectral density (PSD) was calculated using fast Fourier transformation (FFT) as a density spectral array for the following spectral bands: 3–4 Hz, 4–5 Hz, 5–6 Hz, 6–7 Hz, … 19–20 Hz). For further analysis, we used log-transformed values. 

The fractal dimension (FD) was calculated from the signal bandpass filter in the range of interest (2–20 Hz) with a Butterworth filter of the order 12. The fractal dimension (FD) was evaluated using the Higuchi algorithm. 

The Hjorth complexity (HC) parameter, which represents the change in frequency and indicates how the shape of a signal is similar to a pure sine wave, was calculated for a wideband 1.6–30 Hz filtered signal in the following way: complexity(y(t))=, where mobility(y(t))=, y(t)-a signal, y’(t)-its derivative, and var(…)-the variance.

### 2.3. Literature Search for Scoping Review 

To investigate the occurrence of ASD among patients affected by Potocki–Lupski syndrome and to analyze the previous literature, we conducted a systematic search through the PubMed database. For searching relevant articles, the following search terms were used: (Potocki–Lupski) AND (autis * OR ASD), applying to titles and abstracts for studies published in English. The search was performed on 8 November 2022. In addition, articles from a previous literature review [2] were included if they were not identified through PubMed. The exclusion criteria were the following: (1) not-full text articles (such as letters and conference theses); (2) studies without original data (i.e., different types of reviews and meta-analyses); (3) animal studies; (4) studies based on group comparison designs; and (5) the age of participants lower than 18 months (the minimal age for a reliable diagnostic of ASD [21]).

## 3. Results

### 3.1. Clinical and Behavioral Assessment

#### 3.1.1. Case Presentation

The child has no siblings. Pregnancy was without any complications. No infections, medication, smoking, or intake of alcohol or drugs during pregnancy were reported. At the time of the child’s birth, the mother and father were 34 and 30 years of age, respectively. No genetic syndromes have been reported in the family. The patient was delivered in week 42 of gestation as a result of artificially induced labor because the fetus had an abnormally slow heartbeat and excess fluid in the lungs. Thus, after delivery, she was placed in a neonatal intensive care unit. Birth weight was 2920 g (25th percentile). From the first months of life, the child struggled with multiple problems: there was a failure to thrive, poor feeding with a lack of sucking and vomiting after feeding, and sleep disturbances. In early childhood, the patient also had a history of breath-holding spells (with loss of consciousness) and several episodes of fever greater than 40 °C with febrile seizures. Her gross motor milestones were met at the late end of normal limits. She achieved walking at 14 months, but for a long time, she did it awkwardly. Fine motor skills remained challenged for a while. Bowel and bladder control were delayed and obtained after 40 months. In the process of language development, she had no history of babbling, and her first vocalizations were reported to be like “whistling”. The patient’s first words appeared at about 18 months but were sporadic, and she did not say relatively stable words until three years of age and did not begin to generate simple three to five word phrases until four years old. Additionally, remarkable delays were reported in receptive language: at the age of four to five, she understood about 50 words and did not follow complex requests, so her parents tried to break down instructions into minimal blocks. Reportedly, it seemed that she was mostly guided by the context instead of the meaning of receptive phrases. At the same time, her own speech was also poorly understood by others because of intonational problems and a lack of lexical skills. By the age of five, the patient was diagnosed with developmental delay, sensory-motor alalia, ADHD, speech and language impairments, and a number of learning disorders (dyslexia, dysgraphia, and dyscalculia). Her mother also mentioned that clinicians, who assessed the patient, stated that the child had traits of ASD. Initially, the patient was referred to the research team at 11 years of age for clinical evaluation, which revealed delays in speech and cognitive development and problems in adaptive functioning, and concerns about the genetic underpinnings of the disorders. Later, at the age of 13, the patient was recommended to undergo genetic testing [6]. Genome screening of the family trio was carried out using whole-exome sequencing (WES). The analyses revealed multiple de novo single nucleotide variants (SNVs) and copy-number variations (CNVs), two of which, the deletion 15q11.2 and the duplication 17p11.2, have clinical significance. Molecular cytogenetic testing using the FISH technique did not confirm 15q11.2 deletion. In contrast, 17p11.2 duplication was confirmed; thus, the patient was diagnosed with Potocki–Lupski syndrome. The patient’s clinical presentation corresponds highly to the main features of the syndrome (i.e., multiple developmental delays, muscle hypotonia, feeding problems, and behavioral disorders).

Up to the current survey performed at the age of 13, the patient attended a regular public school, mastering an individual educational plan in inclusive settings (mainly aimed at children with ASD). Her parents noticed that she still had difficulties with comprehension of addressed speech and problems with pronunciation, intonation, and fluency of speech. Her academic skills were described as lagging: she had only recently begun to master writing and had difficulties with arithmetic and memorizing. The patient’s mother described her as very sociable, friendly, and gullible and said that she is very attracted to people, has high empathy, and loves to play with peers (especially in story and role-playing games).

#### 3.1.2. Core Features of Autism Spectrum Disorder

The results of the ADI-R were above the clinical cut-off for ASD only for the communication domain (B = 9, cut-off = 8) and for the age of the manifestation of the symptoms (D = 5, cut-off = 1). 

At the beginning of the ADOS-2 assessment, the patient was embarrassed. She supported the role play but did not take the initiative herself, becoming more active during the conversation and discussion of various topics within Module-3. At the same time, throughout the assessment, she maintained eye contact and smiled a lot. As a result of the evaluation (see Table 1), she endorsed six points on the clinical scale (standardized score-three), which corresponds to the ADOS classification outside the autism spectrum.

#### 3.1.3. Language Development

The patient’s performance on the RPLS-5 was substantially lower than that of her peers. Her raw scores were 56 out of 65 for Auditory Comprehension and 54 out of 67 for the Expressive Communication subscales. Her total language ability in Russian corresponded to the age equivalent of five years two months (which should be interpreted with caution, given that only English norms are available). In the receptive domain, the patient exhibited difficulties following complex three-step instructions, in the comprehension of some logical operators (such as “before”-“then”), understanding prefixes, rhymes, and sound composition of words, and in answering questions on story comprehension. In expressive communication, the patient was able to build complex sentences of four to five words and talk about her toys and important life events, except for situations needed to explain the use of objects, reasons, and consequences. Results of the ORRIA assessment indicated that the most evident weaknesses were found in sentence repetition: as the number and complexity of tasks increased, she more often repeated only the last word in the sentence. Furthermore, she had difficulties in items indicating working memory abilities and mastery of complex semantic structures with logical, temporal, and spatial relationships. 

#### 3.1.4. Cognitive Development

The intelligence quotient of the patient, as assessed by the UNIT-2, comprised 64; she was ranked at the first percentile. As seen in Table 2, this result was mostly influenced by low scores in the Reasoning composite, which was a relative weakness for her, indicating that she performed particularly poorly on tests that required pattern processing, awareness of visual–spatial mappings, and understanding of geometric relationships.

#### 3.1.5. Adaptive Functioning

According to Vineland-II results, the patient (Table 3) demonstrated moderately low and low skills on all indicators of adaptive functioning, except for receptive language, in which she had adequate development. The most profound area of concern was identified in the written communication domain. The overall analysis of her profile showed low adaptive functioning with mild deficits in the majority of developmental areas.

### 3.2. Electroencephalographic Assessment

None of the 37 control peers had episodic peak–wave or slow spike–wave complexes or typical or atypical epileptiform discharges. The patient demonstrated atypical and typical paroxysmal activity with a total duration of 292.3 s. (16.65% from the analyzed EEG fragment (1756 s.)). The series of atypical paroxysmal slow spike-and-slow wave discharges accounted for 43.6% of the total paroxysmal activity, appeared 22 times, and had a mean duration of 5.8 ± 2.69 s. and a mean frequency of 2.38 ± 0.48 Hz. At the same time, the appearance of these slow spike-and-slow wave discharges in 9 out of 22 times was accompanied by the loss of vocal activity and atypical face movements. The series of saw-wave patterns accounted for 31.7% of the total paroxysmal activity, appeared 29 times, and had a mean duration of 3.2 ± 1.88 s. and a mean frequency of 13.3 ± 0.15 Hz (Figure 1). The rest of the benign paroxysmal activity consisted of non-epileptic paroxysmal events, including single or series of spikes, charges, and sharp waves. 

Conventional quantitative analysis of EEG PSD did not reveal any differences between the patient from typically developing peers: all values lie within 1 SD from the mean. At the same time, non-linear features of EEG were abnormal. She had considerably lower fractal dimension and Hjorth complexity compared to healthy peers (see Figure 2). Moreover, all children from the control group had higher values of FD and HC compared to the patient (<4σ).

### 3.3. Scoping Review

The initial search identified 30 citations from PubMed and 11 from the reference list of the previous literature review [2]. We retrieved 33 full-text articles assessed against inclusion and exclusion criteria, after eight citations were removed as duplicates. Thus, 14 articles were evaluated as eligible for the current analysis. Their summary characteristics concerning the occurrence of ASD or autistic traits and other neurodevelopment or behavioral and affective disturbances among reported cases are presented in Appendix A. The retrieved sample (including the patient from the current study) comprised 53 individuals (27 females) from 18 months to 47 years of age. Among them, 11 (20.7%) individuals were identified as having ASD, and 13 (24.5%) as having autistic features. However, only one study mentioned the diagnostic criteria used for case evaluation [22], and three studies used standardized comprehensive assessments of ASD manifestations (i.e., ADOS and ADI-R) [1,2,4]. Most patients with PTLS had a history of developmental delay (88.7%), and 25 cases (72.1%) were described as having an intellectual disability. Different types of speech and language impairments were also common and noticed in 27 cases (60%). Other clinical and behavioral disturbances noted among patients with PTLS included hyperactivity and ADHD features, problems with executive functions, aggressiveness, anxiety, obsessive–compulsive behaviors, withdrawal, and learning disorders. In one case, PTLS was also accompanied by bipolar affective disorder [23].

## 4. Discussion

Here we present a case of a female with PTLS focusing on in-depth clinical, behavioral, and electrophysiological evaluation, specifically concerning the features and correlates of ASD, which is commonly reported as one of the most prevalent disorders, that could describe the behavioral features of this syndrome [1,2,4]. Thus, the reported occurrence of ASD among PTLS patients ranges from 37.9 to 80% [1,2,24]. This range seems to be substantially larger than for most other common genetic syndromes, for whom the prevalence of ASD ranges from 11 to 61%, as was found in the systematic review and meta-analysis published in 2015 [25]). However, there are reports of individuals with PTLS who do not reveal autistic features or who demonstrate some of them, not fitting into the whole clinical picture of ASD (Table A1). Thus, in the study of Ercan-Sencicek et al. [5], the central role of autism-associated features has been questioned. Instead, this key phenotype role was offered to speech and language disorders. Thus, it can be assumed that communication difficulties in patients with PTLS could not be a manifestation of “true” ASD but rather be related to language impairments. 

Our scoping review of PTLS reports reveals that a history of developmental delay has been observed in most patients with PTLS. Among behavioral and psychiatric diagnoses, intellectual disability was relatively the most common (88.7% of the reported cases), followed by different types of speech and language impairments (60%), and then ASD (20.7%). Thus, the obtained preliminary occurrence of ASD in PTLS is substantially higher in comparison with ~1% in the general population [26]. However, only one study mentioned the diagnostic criteria used for ASD case evaluation [22], and only three studies used standardized comprehensive assessments of ASD manifestations (i.e., ADOS and ADI-R) [1,2,4], which brings into question the reliability of the obtained estimate for ASD. The obtained estimate represents the lowest value compared with previous reports. Concurrently, 24.5% of the reported cases identified in the literature were also found to have some autistic features. Accordingly, we can assume that equating the full clinical picture of ASD and some of its features could lead to an overestimation of the ASD occurrence among patients with PTLS. 

The case presented in the current study supports the hypothesis that speech and language disorders play a more central role in the PTLS phenotype than ASD. Although mild communication impairments were revealed, obtained results had not fulfilled both diagnostic domains of ASD [19]. At the same time, multiple alterations in language, both expressive and receptive, were identified, along with mild intellectual disability that is also reported to be prevalent among patients with PTLS [1]. 

Although we suppose that the prevalence of ASD in patients with PTLS is likely overestimated, we nevertheless suggest that this syndrome should be considered in the course of etiologic workup following ASD diagnosis confirmation, according to stepwise clinical guidelines recommended by the American Academy of Pediatrics [21]. 

As previous findings demonstrated that EEG abnormalities were more common in children with genetic syndromes associated with autism compared to typically developing children [27,28], we also provided results of the EEG examination of our patient. As for the PTLS, sporadic paroxysmal EEG abnormalities without clinical correlates were reported in 12–45% of cases, and none of the subjects demonstrated seizure discharges [3,4,28,29]. During the clinical EEG analysis of the current case, we revealed two types of atypical paroxysmal EEG abnormalities, which were not previously revealed in patients with the same pathology. Firstly, we have found the atypical peak–slow wave patterns that were previously found in girls with Rett syndrome associated with the later onset of the disease [30]. The clinical correlates of the described atypical peak–slow wave patterns are still under discussion due to a progressive loss of motor skills in children with Rett syndrome and the atypical facial movements that were observed in our patient. At the same time, the atypical facial movements looking like a grimace could be mostly associated with the clinical correlates of this abnormal EEG pattern. Secondly, the saw-like sharp waves with a frequency of 13 Hz registered in our patient were not identified in the literature (in ASD-associated syndromes in particular and neurodevelopmental disorders in general) known to us (however, a more comprehensive and systematic investigation of this issue in the literature is needed).

The analysis of the power spectral density of the resting state EEG using PSD did not show considerable differences between the present patient and the cohort of her healthy peers. At the same time, the values of non-linear features such as HC and FD were noticeably lower in our patient. The Fourier transform method is a common technique of EEG analysis; however, it is poorly suited for the analysis of non-stationary departures of the EEG signal. At the same time, the Hjorth parameters and FD were previously used for the analysis of the abnormal activity of the EEG [31,32,33]. 

Further, these parameters were sensitive to pathological states, including neurological and psychiatric diseases. In particular, the Hjorth complexity and fractal dimension of the EEG were significantly lower compared to healthy subjects in comatose patients and patients with ischemia [34], and the Hjorth complexity was reduced in children with ASD compared to their typical peers [35]. Therefore, the considerably lower values of the non-linear features of the EEG could be a sign of both non-specific neurological or mental pathology and specific EEG dynamics of PTLS, thus this issue should be further investigated.

Finally, it is worth noting that this study has a number of limitations. As for the case report, given the heterogeneity of the clinical manifestations of PTLS, the findings could be unrepresentative for this population. Moreover, considering the scoping review, it should be noted that due to the wide age range and inconsistency among diagnostic methods and sample sizes of the included reports, all obtained results should be regarded as preliminary and interpreted with caution.

## 5. Conclusions

The presented case contributes to the data on phenotypic presentations of the relatively rare and newly recognized genetic condition PTLS. Although PTLS is commonly reported to be accompanied by the clinical picture of ASD, we have not identified this disorder in our patient, mostly characterized by cognitive delay and language impairments. The analysis of clinical EEG has found atypical peak–slow wave patterns and a unique saw-like sharp wave of 13 Hz that was not identified in the previous literature known to us. At the same time, the analysis of the power spectral density of the resting state EEG using PSD did not show considerable differences between the presented case and the cohort of healthy peers. However, the values of non-linear features, such as Hjorth complexity and fractal dimension, were noticeably lower in our patient compared with her neurotypical peers. The scoping review of the identified literature concerning cases with PTLS demonstrates that among reported cases, the most common behavioral and psychiatric diagnoses were intellectual disability, followed by different types of speech and language impairments, and then ASD.

## Figures and Tables

**Figure 1 jpm-13-00439-f001:**
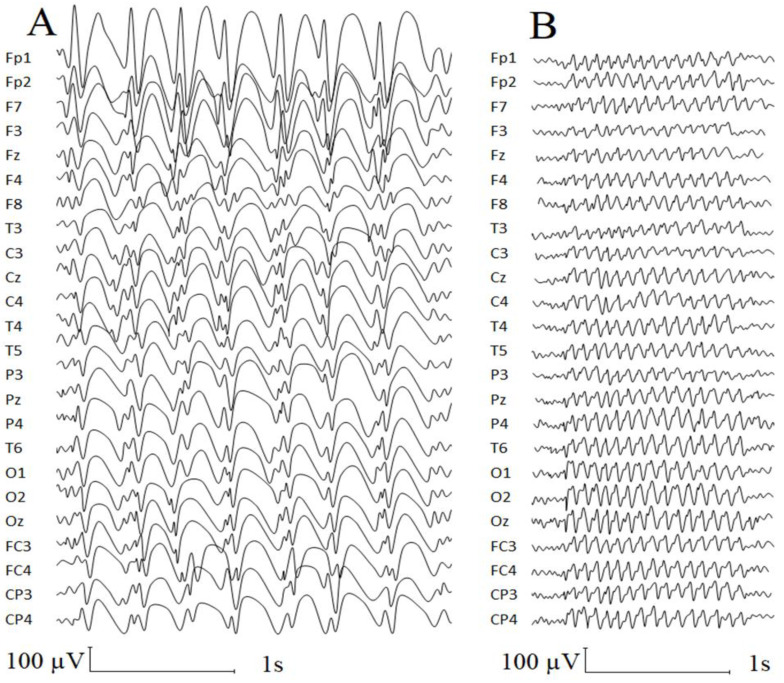
The types of atypical paroxysmal activity found in the current patient: (**A**) atypical absences; (**B**) saw-wave patterns (13.3 Hz). We used base-monopolar montage. A1 and A2 reference electrodes were located on the earlobes.

**Figure 2 jpm-13-00439-f002:**
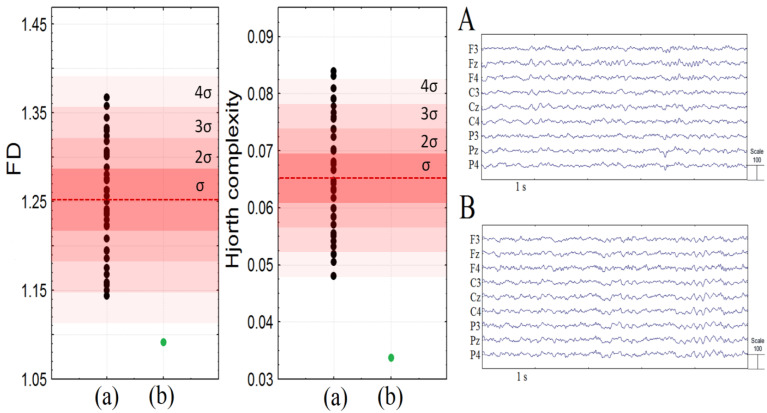
The fractal dimension (FD) and Hjorth complexity over central, frontal, and parietal areas in children of the control group (a) and Patient M (b). (**A**): raw EEG screenshot of a healthy 14 y.o. girl from the control group; (**B**): the raw EEG screenshot of Patient M. The red areas depict the 1–4 sigmas, and the dotted red line is the mean.

**Table 1 jpm-13-00439-t001:** Results of the Assessment of Core Symptoms of Autism using ADOS-2 (Module 3).

ADOS-2 Scales	Item	Score
Social affect (SA)		
*Language and communication*		
Reporting of events	A-7	2
Conversation	A-8	1
Descriptive, conventional, or informational gestures	A-9	1
*Reciprocal social interaction*		
Unusual eye contact	B-1	0
Facial expressions directed to others	B-2	0
Shared enjoyment in interaction	B-4	1
Quality of social overtures	B-7	1
Quality of social response	B-9	0
Amount of reciprocal social communication	B-10	0
Overall quality of rapport	B-11	0
*SA total*		*6*
Restricted and repetitive behavior (RRB)		
*Play, stereotyped behaviors and restricted interests*		
Stereotyped/idiosyncratic use of words or phrases	A-4	0
Unusual sensory interest in play material/person	D-1	0
Hand and finger and other complex mannerisms	D-2	0
Excessive interest in or references to unusual or highly specific topics or objects or repetitive behaviors	D-4	0
*RRB total*		*0*
*Overall total*		*6*

**Table 2 jpm-13-00439-t002:** Results of Intellectual Functioning using UNIT-2.

UNIT-2 Composite	Index Score	95% CI	Percentile Rank	Descriptive Classification
Memory	73	[67, 82]	4	Delayed
Reasoning	65	[61, 72]	1	Very delayed
Quantitative	71	[67, 77]	3	Delayed
Full scale battery	64	[61, 69]	1	Very delayed

**Table 3 jpm-13-00439-t003:** Results of the Adaptive Functioning Assessment using Vineland-II.

Domain	Scores	V-Scores	Descriptive Classification	Standard Scores	Percentile
Receptive	39	14	Adequate		
Expressive	81	7	Low		
Written	19	7	Low		
*Communication*		28	Low, mild deficit	69	2%
Personal	66	9	Low		
Domestic	24	10	Moderately low, mild deficit		
Community	29	7	Low		
*Daily living skills*		26	Low, mild deficit	65	1%
Interpersonal relationship	68	10	Moderately low		
Play and leisure time	44	8	Low		
Coping skills	39	12	Low		
*Socialization*		30	Moderately low	73	4%
** *Adaptive behavior composite* **			Low, mild deficit	67	1%
Internalizing	4	18	Elevated		
Externalizing	1	16	Average		
Maladaptive behavior index	9	17	Average		

## Data Availability

No new data were created or analyzed in this study. Data sharing is not applicable to this article.

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
