# Peer review of "Does the Potocki–Lupski Syndrome Convey the Autism Spectrum Disorder Phenotype? Case Report and Scoping Review"

_jpm, 2023, doi:10.3390/jpm13030439_

Round 1

Reviewer 1 Report

It is an interesting paper presenting the relatively rare and newly recognized genetic condition PTLS and it could be informative for clinicians. The authors nicely questioned the clinical picture of ASD and connection with PTLS. The analysis of clinical EEG has found atypical peak-slow wave patterns and a unique saw-like sharp wave of 13 Hz that was not identified in previous literature. They also present a nice literature review relevant to the discussion. I will suggest to add in case report about the sibling, also about the number of febrile seizures and to disk this relevance in the paper. Also, I would suggest the shortness of 3.1 sections I do not find the relevance to go so deeply into an analysis of the clinical assessment. For the EEG part and literature review I will stay as it is. Also, I will suggest comparison with other neurdevelopmental disorders that might be relevant to discuss regarding EEG abnormalities. At the end I will suggest to add some recommendations – do we need genetic screening in all neurodevelopmental disorders, the relevance of that suggestion and similar

Author Response

Point 1: I will suggest to add in case report about the sibling, also about the number of febrile seizures and to disk this relevance in the paper.

Response 1: Information about siblings has been added. As for febrile seizures, according to the information obtained during the clinical interview, the presented patient had several episodes (added in the text) of febrile seizures (a more precise number was not obtained).

Point 2: Also, I would suggest the shortness of 3.1 sections – I do not find the relevance to go so deeply into an analysis of the clinical assessment.

Response 2: We have shortened this section slightly, but it seems essential to us to retain a more detailed description of the behavioral characteristics of the presented patient because (1) in-depth clinical evaluation is one of the goals of this article; (2) a reliable diagnostics of ASD in case of some rare genetic disorders (like PTLS) with complex and mosaic clinical picture may be a challenging task, so it seems important to us to describe the results transparently; and (3) the number of reported PTLS cases is still limited, while the issue about behavioral phenotype is currently under discussion, we suppose that detailed reports may be vital for accumulating data and formulating ideas for further research.

Point 3: For the EEG part and literature review I will stay as it is. Also, I will suggest comparison with other neurdevelopmental disorders that might be relevant to discuss regarding EEG abnormalities.

Response 3: As we did not identify additional literature relevant to compare revealed abnormalities with those reported for other neurodevelopmental disorders, we replaced the sentence “Secondly, the saw-like sharp waves with a frequency of 13 Hz found in our patient were not identified in the literature (on ASD-associated syndromes) known to us” in lines 321-325 by “Secondly, the saw-like sharp waves with a frequency of 13 Hz registered in our patient were not identified in the literature (in ASD-associated syndromes in particular and neurodevelopmental disorders in general) known to us (but more comprehensive and systematic investigation of this issue in the literature is needed).”

Point 4: At the end I will suggest to add some recommendations – do we need genetic screening in all neurodevelopmental disorders, the relevance of that suggestion and similar

Response 4: The following statement was added in the Discussion: “Although we suppose that the prevalence of ASD in patients with PTLS is likely overestimated, we nevertheless suggest that this syndrome should be considered in the course of etiologic workup following ASD diagnosis confirmation according to stepwise clinical guidelines recommended by the American Academy of Pediatrics [21]”.

Reviewer 2 Report

The authors described a case of a female adolescent with a de novo dup(17)(p11.2p11.2) without ASD features, focusing on in-depth clinical, behavioral, and electrophysiological evaluations. There are no previous case reports which figure out detailed clinical and neurodevelopmental characteristics, therefore the present study is valuable.

I have some concerns about this report.

Major points

1)    Genetic test for diagnosing Potocki-Lupski syndrome

The authors described that she had a de novo dup(17)(p11.2p11.2) in the Case presentation section (reference number [6]). I cannot read the cited study [6], therefore I cannot judge the validity of genetic test performed in the present patient. Especially, I want to know whether she had other mutations responsible for her neurodevelopmental or electroencephalogram findings. Did the authors check comprehensive gene analysis using next-generation sequencer in the present patient?

2)    Ethical aspect

I cannot confirm the ethical statements in the present study. Did the authors obtain informed consent from the present girl? Were the normal controls of electroencephalogram which were registered for other study acceptable in the present study through an appropriate ethical procedure?

Minor points

3)    Address term

The present case should be addressed as not “the Patient M” but “the present patient”, for example.

4)    Electroencephalogram

The authors should present the reference electrode in the Figure 1.

Author Response

Point 1: The authors described that she had a de novo dup(17)(p11.2p11.2) in the Case presentation section (reference number [6]). I cannot read the cited study [6], therefore I cannot judge the validity of genetic test performed in the present patient. Especially, I want to know whether she had other mutations responsible for her neurodevelopmental or electroencephalogram findings. Did the authors check comprehensive gene analysis using next-generation sequencer in the present patient?

Response 1: We are surprised to learn that reference #6 is not easily available to the reviewer; we checked and had no difficulty finding the article through its DOI. Yet, although constrained by space, we included critical details of the molecular procedures in the manuscript.

Point 2: I cannot confirm the ethical statements in the present study. Did the authors obtain informed consent from the present girl? Were the normal controls of electroencephalogram which were registered for other study acceptable in the present study through an appropriate ethical procedure?

Response 2: Thanks for this comment! We have corrected our ethical statements and hope that they become clearer (see sections “Institutional Review Board Statement” and “Informed Consent Statement”). We have also found that we had not previously indicated that two forms of informed consents were used for the present patient, approved by two ethics committees: for behavioral and EEG examinations separately (so we have added information about this).

Point 3: The present case should be addressed as not “the Patient M” but “the present patient”, for example.

Response 3: Thank you for this suggestion! We have made corresponding edits.

Point 4: The authors should present the reference electrode in the Figure 1.

Response 4: Thank you! Information about reference electrodes was added to the notes for Figure 1.

Round 2

Reviewer 2 Report

The re-submitted manuscript was revised adequately.